# Seroprevalence of HIV, HBV, and syphilis co-infections and associated factors among pregnant women attending antenatal care in Amhara regional state, northern Ethiopia: A hospital-based cross-sectional study

**Degsew Ewunetie Anteneh**[1]*, **Eden Bishaw Taye**[2], **Asmra Tesfahun Seyoum**[2], **Alemken Eyayu Abuhay**[3], **Endeshaw Admassu cherkose**[2]

1 Department of Clinical Midwifery, School of Midwifery, College of Medicine and Health Science, Woldia University, Woldia, Ethiopia, 2 Department of Clinical Midwifery, School of Midwifery, College of Medicine and Health Science, University of Gondar, Gondar, Ethiopia, 3 Department of Clinical Midwifery, University of Gondar Comprehensive Specialized Hospital, Gondar, Ethiopia

* degsewewunetie@gmail.com

## Abstract

### Background

Co-infections involving human immunodeficiency virus (HIV), hepatitis B virus (HBV), and syphilis pose significant public health problems during pregnancy. It can increase the risk of adverse outcomes for both the woman and the infant more than each infection alone does. However, the magnitude of these co-infections remains insufficiently documented. Hence, this study aimed to determine the seroprevalence of HIV, HBV, and syphilis co-infections and associated risk factors among pregnant women attending antenatal care in Amhara region referral hospitals in northern Ethiopia.

### Methods

A hospital-based cross-sectional study was conducted in Amhara regional state referral hospitals from January 1 to February 30, 2024, among 606 pregnant women. Pregnant women were selected using a systematic random sampling technique. An interviewer-administered questionnaire and chart review were used to collect data. Data were analyzed in SPSSV26.0. Descriptive statistics were used to determine the magnitude of co-infections, and binary logistic regression was used to determine associated factors. Variables with a P-value < 0.05 were used to declare statistical significance.

### Result

Overall, 4.1% (95% CI: 2.7, 6.1) of pregnant women were co-infected. The prevalence of specific co-infections was 2% (95% CI: 1, 3.5) for HIV/HBV, 1.3% (95% CI: 0.6, 2.6) for HIV/syphilis, and 0.8% (95% CI: 0.3, 1.9) for HBV/syphilis. No cases of triple co-infection were observed. Women with a history of unsafe sex (AOR = 8.2, 95% CI: 1.5, 16.7) and

**Data Availability Statement:** All relevant data are within the paper and its Supporting Information files.

**Funding:** The author(s) received no specific funding for this work.

**Competing interests:** the authors have declared that no competing interests exist

**Abbreviations:** ANC, Antenatal Care; AOR, Adjusted Odds Ratio; CI, Confidence Interval; COR, Crude Odds Ratio; HBsAg, Hepatitis B Surface Antigen; HBV, Hepatitis B Virus; HIV, Human Immunodeficiency Virus; MTCT, Mother to Child Transmission; PMTCT, Prevention of Mother-to-Child Transmission; SPSS, Statistical Product and Service Solutions; STIs, Sexually Transmitted Infections; WHO, World Health Organization.

incarceration (AOR = 9.3, 95% CI: 1.6, 20.8) were associated with HIV/syphilis co-infection. For HIV/HBV co-infection, contact with jaundice patients (AOR = 5.5, 95% CI: 1.3, 22.5) and women with a history of STIs (AOR = 4.6, 95% CI: 1.4, 14.9) was significantly associated. Women with STI history (AOR = 6.3, 95% CI: 1.2, 15.9) were also significantly associated with HBV/syphilis co-infection.

## Conclusion

Despite the government's elimination efforts, a relatively high prevalence of coinfections with the infections studied was found among pregnant women. Therefore, HIV, HBV, and syphilis testing and treatment packages should be strengthened by targeting pregnant women with a history of STIs, contact with patients with jaundice, a history of incarceration, and unsafe sex.

## Background

Globally, human immunodeficiency virus (HIV), hepatitis B virus (HBV), and syphilis pose a significant public health challenge to pregnant women due to vertical transmission [1]. An estimated 1.3 million women become pregnant with HIV each year, and approximately 470,000 children become infected with HIV through mother-to-child transmission (MTCT). It leads to acquired immunodeficiency syndrome (AIDS) and affects pregnancy outcomes, including intrauterine growth restriction (20.5%), preterm birth (25.0%), and cesarean delivery (45.5%) [2]. Approximately 5–20% of these women were found to be coinfected with hepatitis B virus (HBV) [3, 4].

Syphilis is a chronic infectious disease caused by *Treponema pallidum* that causes ulcerative genital lesions and increases sexually transmitted infections [1]. It affects around 900,000 pregnant women worldwide each year, with the highest prevalence in Africa (39.3%) and Asia (44.3%) [3]. Untreated cases can lead to adverse outcomes in 5–8% of pregnancies [5]. Pregnant women may experience serious congenital infections, leading to fetal or neonatal death (50%), premature birth (25%), and serious long-term sequelae in surviving children (20%) [2].

Hepatitis B virus, a double-stranded deoxyribonucleic acid virus that primarily affects the liver, is the sixth leading cause of death worldwide [6]. This results in significant mortality due to acute and chronic infections. The World Health Organization (WHO) estimates that around 296 million people worldwide suffer from chronic HBV, with 1.5 million new infections and 820,000 deaths per year [3]. Up to 90% of babies born to HBV carriers develop chronic liver disease at a younger age, which only affects about 5% of those infected in adulthood [1, 3].

Due to shared transmission routes and risk factors such as unsafe sexual contact, contaminated blood, and mother-to-child transmission (MTCT), coinfections with HIV, HBV, and syphilis in pregnant women can significantly worsen clinical outcomes and influence the epidemiology of each infection through various mechanisms [7]. These mechanisms include immune-mediated injury, oxidative stress, mitochondrial damage, lipotoxicity, cytotoxicity, toxic metabolite accumulation, and systemic inflammation. Such interactions can increase the risk of mother-to-child transmission, accelerate disease progression, and heighten susceptibility to opportunistic infections, thereby adding stress to the pregnancy [1]. For instance, pregnant women who have both HIV and HBV are more likely to pass on either virus to their

unborn child and are at a higher risk of developing hepatocellular carcinoma and liver cirrhosis, which can increase liver-related morbidity and mortality four to five times higher than in cases of HBV infection alone [2]. Likewise, babies born to women who have both HIV and syphilis are at higher risk of becoming infected with either infection than babies who only have HIV or syphilis, and vice versa [8].

Furthermore, these health challenges extend to broader societal problems, such as poverty, and ultimately threaten community well-being [1, 2, 9]. The problem of these infections is common in developing countries, including Ethiopia, which suffers from poverty and inadequate health care [3]. Studies from different countries have found different rates of coinfection with HIV, HBV, and syphilis, as well as important factors that increase the risk of these diseases. Based on the type of coinfection, the overall prevalence of HBV-HIV coinfection in pregnant women in sub-Saharan Africa is approximately 3.302% [10]. Specific rates vary by region in Africa, with the northern region of Cameroon at 1.5% [11], Nigeria among HIV positive pregnant women (12.6%) [12], Rwanda 4.1% [13], 9.5% in Zambia [14], 6.3% in Angola [15].Regarding HIV/syphilis coinfection; In Botswana, 37.7% of pregnant women were coinfected [16], in Angola 6.3% [17], in Rwanda 27.4% [13] and in our country Ethiopia 2 .2% [18] and 1.1% [9] respectively were coinfected with HIV/syphilis (16), and 2.1% in Yirgalem hospital [19] and 9.5% in Gandhi Memorial Hospital, Addis Ababa, were coinfected with HIV/HBV [20].

To mitigate these pandemics, global public health for the years 2022–2030 under the MTCT has established the triple elimination program for HIV, HBV, and syphilis to reduce pregnancy-related deaths in infected women and eliminate these infections in children [3]. Ethiopia also initiated this program with the aim of achieving specific targets by 2025 [1, 3]. The country has also implemented various national strategies, programs, and guidelines, including syndromic screening and serial testing algorithms for HIV, HBV, and syphilis, along with offering free antiretroviral therapy services [1].

However, different reports indicate that syphilis, HIV, and HBV remain significant public health issues with ongoing coinfection in Ethiopia [1]. Yet information on the disease burden of these coinfections on pregnant women remains unclear. Therefore, this study aimed to determine the seroprevalence of HIV, HBV, and syphilis co-infections and associated risk factors among pregnant women attending antenatal care in Amhara region referral hospitals, northern Ethiopia.

## Methods and materials

### Study design, period, and setting

A multi-center, hospital-based cross-sectional study was conducted at the Amhara regional state referral hospitals from January 1 to February 30, 2024.

The research was conducted in the referral hospitals of the Amhara region in northern Ethiopia. This region is served by eight referral hospitals: University of Gondar Comprehensive Specialized Referral Hospital (UOGCSH), Debre Tabor Comprehensive Specialized Hospital, Debremarkose Comprehensive Specialized Hospital, Woldia Comprehensive Specialized Hospital, Debrebirhan Referral Hospital, Felege Hiwot Comprehensive Specialized Hospital, Tibeb Gion Comprehensive Specialized Hospital, and Dessie Comprehensive Specialized Hospital.

These hospitals have an antenatal clinic where antenatal care services are provided every day. A quite large number of people with different socio-cultural backgrounds visit the hospital every day. According to the 2022–23 report, annually, about 76,470 pregnant mothers visited these hospitals for antennal care services (ANC), including first, second, third, and fourth visits. They serve both local patients and referrals. Their efforts aim to reduce HIV, HBV, and syphilis infections in children and lower pregnancy-related mortality in infected women, in

alignment with the integrated mother-to-child transmission prevention plan set by the Ministry of Health in 2022.

## Source and study population

All pregnant women who were attending antenatal care services at referral hospitals in the Amhara regional state and these pregnant women who were attending antenatal care services in Amhara region referral hospitals during the data collection period were our source and study population, respectively.

## Inclusion and exclusion criteria

All pregnant women attending ANC at the selected referral hospitals were included, and pregnant women who presented with labor were excluded from the study.

## Sample size determination

The sample size was calculated using a single population proportion formula with a 95% confidence level by considering a 9.5% proportion from a previous study of HIV/HBV co-infection in pregnant women in Addis Ababa [20] with a tolerable error of 0.03. The formula used to calculate the sample size was as follows:

$$n = \frac{(z\alpha/2)^2 p(1-p)}{d^2} = \frac{(1.96)^2 0.095(1-0.095)}{0.03^2} = 366.9 \approx 367$$

Where z = Z score for a 95% confidence interval, which is 1.96
p = expected prevalence or proportion of HIV/HBV co-infection which is 9.5% [20].
d = tolerable error between the sample and true population, which is 3%.

After adding a 10% non-response rate to the calculated sample size of 367 and applying a design effect of 1.5, the final sample size was determined to be 606.

## Sampling procedure and technique

A two-stage sampling method was used to select the hospitals and study participants. Initially, referral hospitals were selected, followed by a selection of study subjects. Of the eight existing referral hospitals in the Amhara region, four were selected using a lottery method. These are Woldia comprehensive specialized hospitals, the University of Gondar specialized hospital, Debremarkose comprehensive specialized hospital, and Debre Tabor comprehensive specialized hospital. Next, a proportional allocation of the sample size was used for each selected hospital according to the number of women receiving antenatal care.

The number of participants per month was estimated based on the three-month review of ANC documents from each selected referral hospital. Over a two-month period, the average number of pregnant women attending antenatal care at specific hospitals was the University of Gondar Specialized Referral Hospital (402), Debre Tabor Comprehensive Specialized Hospital (310), Woldia Comprehensive Specialized Hospital (412), and Debre Markose Comprehensive Specialized Hospital (341). The sampling interval is N/n, which is 1465/606 = 2.4≈2. where N is the average total number of pregnant women seeking ANC services at selected referral hospitals over a two-month period. Finally, eligible pregnant women who are receiving antenatal care services at the hospital were recruited by every other pregnant woman (k = 2) for the study until the required sample size was obtained (steps are described in **Fig 1**).

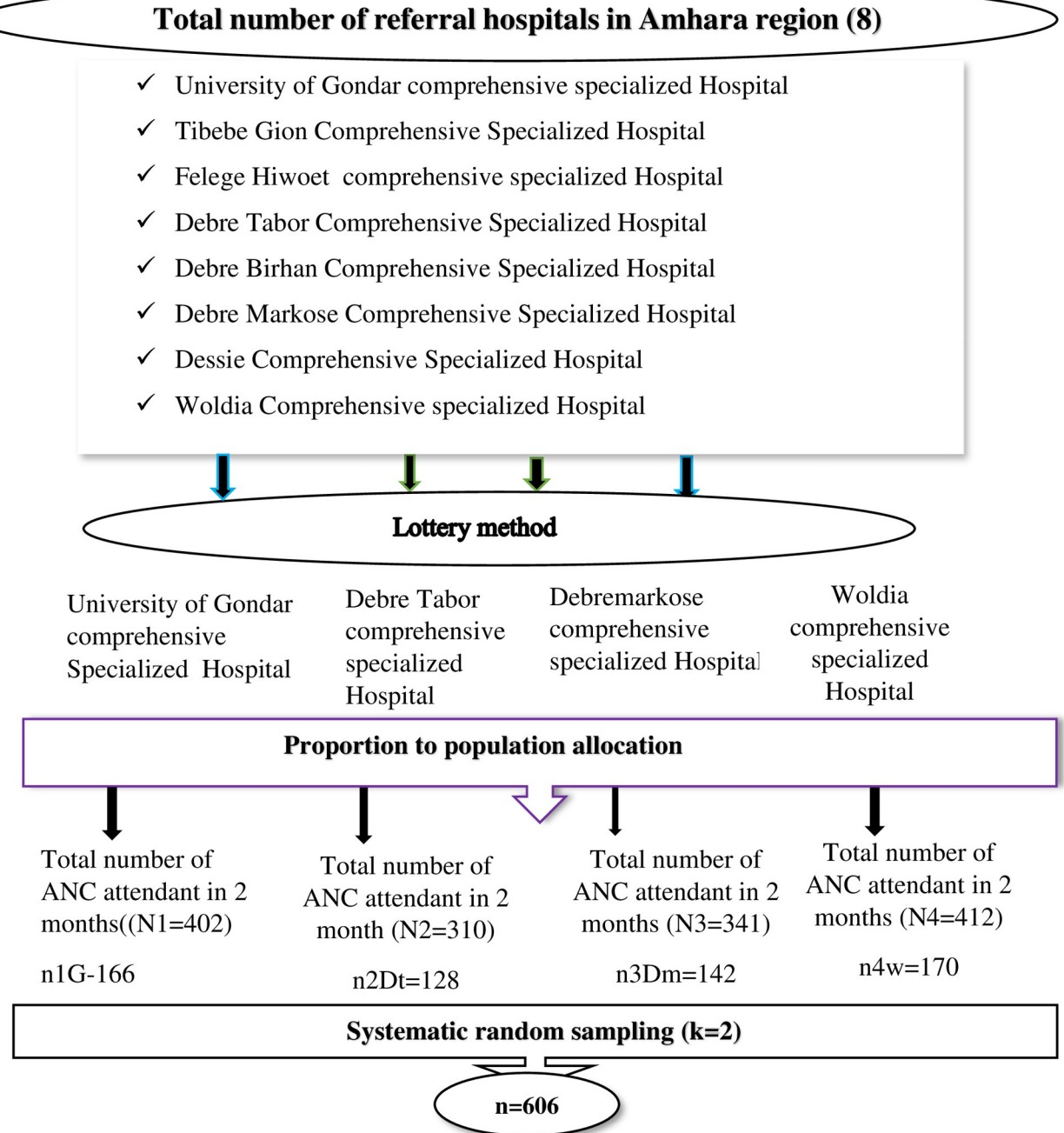

**Fig 1. Schematic diagram showing the sampling technique and procedures to select a sample of 606 pregnant women attending ANC at Amhara region referral hospitals, Ethiopia.**

### Data collection tools and procedures

A structured questionnaire was used to collect the data, which was developed after reviewing various literature [4, 8, 21–23]. Four experienced BSc midwives carried out data collection under the supervision of four MSc midwives. Information regarding socio-demographic factors, maternal-related factors, and risky behavior-related factors was collected from study

participants through face-to-face interviews. To assess the test status for HIV, HBV, and syphilis, chart reviews were conducted by examining medical records to verify the test results for each infection. Since testing for HIV, HBV, and syphilis is routinely performed for all pregnant women during antenatal care (ANC) contacts, we utilized Kobo Toolbox as the data collection platform throughout the process.

## Data quality control

To maintain data quality control, the questionnaire was first prepared in English, translated into Amharic, and then back-translated into English for consistency. Before beginning the actual work, the questionnaire was reviewed and pre-tested on 5% (30) of pregnant women at Tibeb Gion comprehensive specialized hospital. During this pre-test, the questionnaire was checked and corrected for irrelevant and unclear wording. A supervisor (MSc midwives) was assigned to each hospital to ensure that the data was collected properly by data collectors and to address any problems they may have during data collection. A one-day training on the data collection tool and process was conducted for both data collectors and supervisors.

## Laboratory methods

During antenatal care (ANC) visits, all pregnant women routinely undergo screening for HIV, HBV, and syphilis. Laboratory tests are performed using a serological test with five milliliters (5 ml) of blood. Repeat tests are conducted later, after 37 weeks of pregnancy, and based on medical or risky behaviors. Each referral hospital is carried out based on the following methods:

**HIV Testing:** HIV testing is performed using a serial algorithm (three-test algorithm), which includes Test 1 (T1): Stat-pak, Test 2 (T2): Abon, and Test 3 (T3): SD Bioline. The testing procedure is as follows: If the first test (T1) is non-reactive, the final HIV result is negative. If the first test (T1) is reactive, the sample is then tested with the second test (T2). If the second test (T2) is reactive, the sample is further tested with the third test (T3). If the third test (T3) is reactive, the final HIV status is positive, which will then be confirmed at an ART clinic. However, if the first test (T1) is reactive and the second test (T2) is non-reactive, both tests are repeated. After repeating, if T1 remains reactive and T2 is non-reactive, the result is reported as HIV-negative. If both T1 and T2 are non-reactive upon repetition, the result is reported as HIV-negative. If both T1 and T2 are reactive after repeating, the third test (T3) is conducted, and if T3 is reactive, the result is reported as HIV positive.

**Hepatitis B virus (HBV) testing:** Serum samples are tested for HBsAg using the commercially available AiDTM HBsAg enzyme-linked immune sorbent assay (ELISA) test kits developed by Beijing Wantai Biological Pharmacy Enterprise Co., Ltd. To test the presence of HBsAg antibodies in the serum, a Wantai AiDTM HBsAg ELISA test kit has a sensitivity of 100% and a specificity of 99.92%. The serological tests were carried out according to the manufacturer's instructions.

**Syphilis Testing**: Initially, syphilis infection is detected using the Venereal Disease Research Laboratory (VDRL) test, which identifies non-treponemal antibodies. Women who tested positive with the VDRL test are subsequently tested using the immunochromatography test strip (ICS) for the qualitative detection of T. pallidum antibodies. Both tests are conducted according to standard procedures. The ICS test is known for its high sensitivity and specificity.

**Data management and analysis.**   The data exported from Kobo Toolbox was coded, cleaned, stored, and analyzed using SPSS version 26. A statistical summary was performed using descriptive statistics such as frequency counts, percentages, median, and the interquartile range (IQR) to describe the characteristics of pregnant women and estimate the prevalence of

co-infections. It was presented using text, tables, and figures. Binary logistic regression model was done to determine associated factors of co-infections. First, bivariate analysis examined the relationships between each set of independent and dependent variables. Then variables with a p value < 0.25 in the bivariate analysis were considered for the multivariable analysis, controlling for confounding effects. Significant factors of HIV, HBV, and syphilis co-infections were identified based on a p value < 0.05 in multivariable analysis, presented with adjusted odds ratios (AOR) and 95% confidence intervals (CI). Multicollinearity was assessed using the variance inflation factor (VIF) at a cut-off value of 5. Model fitness for multivariable binary logistic regression was checked using the Hosmer and Lemeshow tests. The model was defined to be reasonably well fit at p > 0.05.

**Ethics approval and consent to participate.** Ethical approval was obtained from the school of midwifery on behalf of the institutional review board of the University of Gondar Ethical Review Committee (Mid/W.H./02/16). Formal cooperation letters were sent to each referring hospital. Prior to data collection, every participating mother gave written, informed consent. Participants were informed about the voluntary nature of participation and their right to discontinue at any moment if the questionnaire caused them any discomfort. No personal identifier was included in our data set.

# Result

## Socio-demographic characteristics of pregnant women

Six hundred-three (603) pregnant women participated in the study, resulting in a response rate of 99.5%. Among these women, 568 (94.2%) were married, and the median age of the participants was 31, with an interquartile range (IQR) of 28–33 years. Approximately 40.8% of the participants had attained college education or higher. Regarding employment status and residency, 32% of the participants were employed, and 76.8% lived in urban areas (**Table 1**).

**Table 1. Socio-demographic characteristics of pregnant women attending ANC follow-up at Amhara region referral hospitals, Northern Ethiopia, 2024 (n = 603).**

| Variables | Category | Frequency | Percentage |
|---|---|---|---|
| Age | 18–24 | 42 | 7 |
| | 25–34 | 451 | 74.8 |
| | ≥ 35 | 110 | 18.2 |
| Marital status | Non-married* | 35 | 5.8 |
| | Married | 568 | 94.2 |
| Women's educational status | Unable to read and write | 53 | 8.8 |
| | Primary school | 143 | 23.7 |
| | Secondary school | 161 | 26.7 |
| | Collage and above | 246 | 40.8 |
| Women's occupation | Housewife | 153 | 25.4 |
| | Farmer | 101 | 16.7 |
| | Employed* | 193 | 32 |
| | Private worker | 138 | 22.9 |
| | Student | 18 | 3 |
| Husband's educational status | Unable to write and read | 27 | 4.8 |
| | Primary school | 112 | 19.7 |
| | Secondary school | 137 | 24.1 |
| | Collage and above | 292 | 51.4 |

Employed* = government or non-government employee , Non-married* = divorced or widowed or single

## Maternal, health care, and risky behavior related factors

Among pregnant women across various maternal, health care, and risky behavior-related categories, 53.4% of the women had a history of hospital admissions, and 54.6% of the women were multigravida. Only 1.3% of women received the HBV vaccination; 5.3% of pregnant women had a history of venous or body piercing for treatment; and 5.1% of pregnant women had a history of unsafe sex. Additionally, 4.1% of pregnant women had a history of incarceration (**Table 2**).

## Seroprevalence of HIV, HBV, and syphilis co-infections

The seroprevalence of HIV, HBV, and syphilis infections was 46 (7.6%) (95% CI: 5.6, 10.0), 27 (4.5%) (95% CI: 3.0, 6.4), and 20 (3.3%) (95% CI: 2.0, 5.1), respectively. The co-infection

**Table 2. Some maternal, health care, and risky behavior-related characteristics of pregnant women attending ANC at Amhara region referral hospitals, Ethiopia, 2024 (n = 603).**

| Variables | Category | Frequency, n (%) |
|---|---|---|
| Hospital admission history | Yes | 322 (53.4) |
| | No | 281 (46.6) |
| Surgical procedure | Yes | 104 (17.2) |
| | No | 499 (88.8) |
| Dental procedure | Yes | 99 (16.4) |
| | No | 504 (83.6) |
| Blood transfusion history | Yes | 36 (6) |
| | No | 567 (94) |
| Gravidity | Primigravida | 274 (45.4) |
| | Multigravida | 329 (54.6) |
| Time of ANC initiation | <12 weak | 410 (68.2) |
| | ≥12 weak | 193 (31.8) |
| Current ANC visit | 1st visit | 127 (21.1) |
| | 2nd visit | 257 (42.6) |
| | 3rd visit | 152 (25.2) |
| | 4th and above | 67 (11.1) |
| HBV vaccinated status | Yes | 8 (1.3) |
| | No | 595 (98.7) |
| Pregnancy related complication | Yes | 151 (25) |
| | No | 452 (75) |
| Having a tattoo in the body | Yes | 191 (31.7) |
| | No | 412 (62.3) |
| Use sharp instruments together | Yes | 109 (18.1) |
| | No | 494 (81.9) |
| Having history of unsafe sex | Yes | 31 (5.1) |
| | No | 572 (94.8) |
| History of STI | Yes | 105 (17.4) |
| | No | 498 (82.5) |
| History of incarceration | Yes | 25 (4.1) |
| | No | 578 (95.9) |
| Contact with jaundiced patient | Yes | 37 (6.1) |
| | No | 566 (93.9) |
| History of Venous or body piercing | Yes | 32 (5.3) |
| | No | 571 (94.6) |

prevalence of HIV/HBV was 12 (2%) (95% CI: 1.0, 3.5), HIV/syphilis 8 (1.3%) (95% CI: 0.6, 2.6), and HBV/syphilis 5 (0.8%) (95% CI: 0.3, 1.9). No cases of triple co-infection (HIV/HBV/ syphilis) were observed. A total of 25 (4.1%) (95% CI: 2.7, 6.1) pregnant women were found to be seropositive for more than one infection (**Fig 2**).

### Factors associated with HIV/HBV co-infection

In the bivariate logistic regression analysis, ten (10) variables, such as history of hospital admission, dental procedures, surgical history, gravidity, frequent alcohol consumption, and as listed in Table 3, were found to be candidates for multivariable analysis to the seroprevalence of HIV/HBV co-infection (p ≤ 0.25). However, in multivariable analysis, two variables, pregnant women having a history of contact with jaundiced patients (AOR = 5.5, 95% CI: 1.3, 22.5), and women with a history of STIs (AOR = 4.6, 95% CI: 1.4, 14.9), were associated with HIV/HBV co-infection (**Table 3**).

### Factors associated with HIV/Syphilis co-infection

In examining factors associated with HIV/Syphilis co-infection seroprevalence, the bivariate analysis highlighted nine variables as potential candidates for further investigation (p ≤ 0.25). However, upon conducting multivariable analysis, only two variables, such as history of incarceration (AOR: 9.3, 95% CI: 1.6, 20.8) and a history of unsafe sex among participants (AOR: 8.2, 95% CI: 1.5, 16.7), showed a statistically significant association with HIV/syphilis co-infection (**Table 4**).

### Factors associated with HBV/Syphilis co-infection

In the bivariate analysis, eight variables, including surgical history, husband having another sexual partner, use of sharp material together, unsafe sex history, women with STI history and women having another sexual partner, alcohol consumption, and venous piercing, were considered potential candidates for further analysis. However, in the multivariable analysis, only

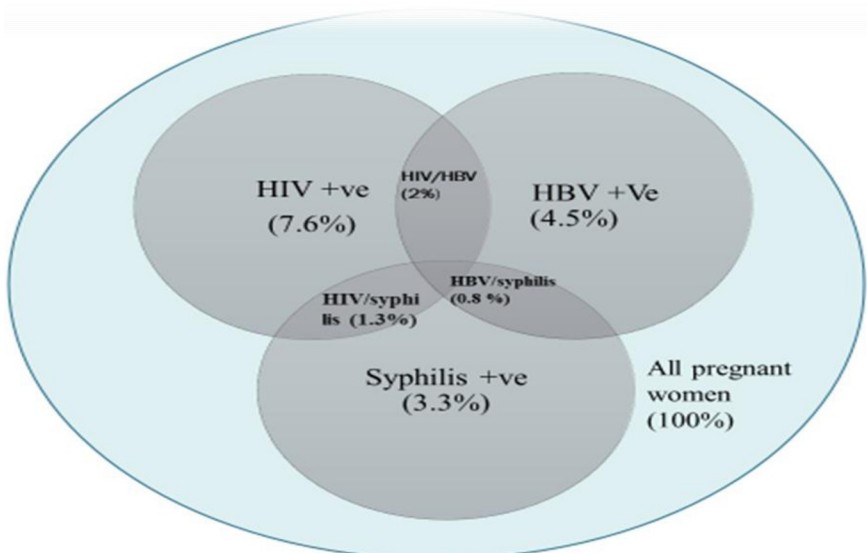

**Fig 2. Prevalence of co-infections with HBV, HIV, and syphilis in 603 pregnant women in Amhara region referral hospitals, Ethiopia.**

**Table 3. Bivariate and multivariable logistic regression analysis for identifying HIV/HBV co-infection risk factors among pregnant women in Amhara region referral hospitals, Ethiopia, 2024 (n = 603).**

| Variables | Category | HIV/HBV | | COR (95% CI) | AOR (95% CI) | P- Value |
|---|---|---|---|---|---|---|
| | | Positive, n (%) | Negative , n (%) | | | |
| Hospital admission | Yes | 10 (3.1) | 312 (96.9) | 4.4 (0.9, 20.5) | 2.1 (0.3, 9.9) | 0.4 |
| | No | 2 (1.1) | 179 (98.9) | 1 | 1 | - |
| Dental procedure | Yes | 4 (4.0) | 95 (96.0) | 2.6 (0.7, 8.8) | 1.4 (0.2, 7.3) | 0.6 |
| | No | 8 (1.6) | 496 (98.4) | 1 | 1 | - |
| Surgical history | Yes | 5 (4.8) | 99 (95.2) | 3.5 (1.1, 11.4) | 1.5 (0.3, 6.4) | 0.6 |
| | No | 7 (1.4) | 492 (98.6) | 1 | 1 | - |
| Gravidity | Primigravida | 2 (0.7) | 272 (99.3) | 0.2 (0.05, 1.1) | 0.7 (0.1, 6.1) | 0.8 |
| | Multigravida | 10 (3.0) | 319 (97.0) | 1 | 1 | - |
| Unsafe sex | Yes | 2 (6.5) | 29 (93.5) | 3.8 (0.8, 18.5) | 1.7 (0.2, 9.5) | 0.6 |
| | No | 10 (1.8) | 562 (98.2) | 1 | 1 | - |
| Women had another sexual partner | Yes | 5 (4.8) | 100 (95.2) | 3.5 (1.1, 11.2) | 1.2 (0.3, 5.9) | 0.7 |
| | No | 7 (1.4) | 491 (98.6) | 1 | 1 | - |
| Women with STI history | Yes | 6 (5.7) | 99 (94.3.) | 4.9 (1.5, 15.7) | 4.6 (1.4, 14.9) | 0.01* |
| | No | 6 (1.2) | 492 (98.8) | 1 | 1 | - |
| History of incarceration | Yes | 2 (8.0) | 23 (92.0) | 4.9 (1.1, 23.8) | 2.6 (0.4, 18.9) | 0.3 |
| | No | 10 (1.9 | 512 (98.1) | 1 | 1 | - |
| Alcohol consumption | Yes | 5 (4.8) | 99 (95.2) | 3.5 (1.1, 11.4) | 2.9 (0.7, 8.8) | 0.1 |
| | No | 7 (1.4) | 492 (98.8) | 1 | 1 | - |
| Contact with jaundiced patient | Yes | 3 (8.1) | 34 (91.9) | 5.5 (1.4, 21) | 5.5 (1.3, 22.5) | 0.02* |
| | No | 9 (1.6) | 557 (98.4) | 1 | 1 | - |

1: reference category; AOR: Adjusted odds ratio; COR: Crude odds ratio; HBV: Hepatitis B virus; STI: Sexually transmitted infection

* The observed difference is statistically significant

women with a history of STI (AOR: 6.3, 95% CI: 1.2, 15.9) showed a statistically significant association with HBV/syphilis co-infection (**Table 5**).

## Discussion

Our study has shown up-to-date evidence about HIV, HBV, and syphilis co-infections among pregnant women in the Amhara region of Ethiopia. The study also revealed factors associated with these co-infections among pregnant women. According to our study findings, the seroprevalence of HBV and HIV co-infection was 2% (95% CI: 1, 3.5). This prevalence of co-infection is comparable to that reported in studies carried out at Yirgalem Hospital in Ethiopia (2.1%) [19], the north region of Cameroon (1.5%) [11], and Sub-Saharan African countries (3.3%) [10]. However, it is lower than the prevalence found in previous studies at Gandhi Memorial Hospital in Addis Ababa (9.5%) [20], and the Ethiopian Population-based HIV Impact Assessment (EPHIA) in the general population is 5.5 [23]. The possible reason for the low prevalence in this study compared to Addis Ababa may be due to higher population density, hotspot areas like bars and brothels, increased migration, and diverse high-risk behaviors in the capital city than in the Amhara region. The lower prevalence of this co-infection among pregnant women compared to the general population may be due to their more frequent medical attention, leading to earlier detection and timely treatment.

It is also lower than the studies conducted outside Ethiopia, such as in Zambia (9.5%) [14], Rwanda (4.1%) [13], Angola (6.5%) [15], Turkey (5.04%) [24], and the studies reported in HIV-positive pregnant women from, Nigeria (12.6%) [12], western China 14.4% [25], and

**Table 4. Bivariate and multivariable logistic regression analysis for identifying risk factors of HIV/Syphilis co-infection among pregnant women in Amhara region referral hospitals, Ethiopia, 2024 (n = 603).**

| Variables | Category | HIV/syphilis | | COR (95% CI) | AOR (95% CI) | P -Value |
|---|---|---|---|---|---|---|
| | | Positive, n (%) | Negative, n (%) | | | |
| Dental procedure | Yes | 4 (4.0) | 95 (96.0) | 5.2 (1.2,21.4) | 5 (0.6, 29.9) | 0.07 |
| | No | 4 (0.8) | 500 (99.2) | 1 | 1 | - |
| Surgical procedure | Yes | 3 (2.9) | 101 (97.1) | 2.9 (0.7,12.4) | 1.4 (0.2,12.2) | 0.7 |
| | No | 5 (1.0) | 494 (99.0) | 1 | 1 | - |
| Having tattoo | Yes | 6 (3.1) | 185 (96.9) | 6.6 (1.3, 33) | 2 (0.3, 15.2) | 0.5 |
| | No | 2 (0.5) | 410 (99.5) | 1 | 1 | - |
| Husband had another sexual partner | Yes | 4 (3.3) | 117 (96.7) | 7.3 (1.3, 40.8) | 1.5 (0.2,15.2) | 0.7 |
| | No | 2 (0.5) | 432 (99.5) | 1 | 1 | - |
| Pregnancy related complications | Yes | 5 (3.3) | 146 (96.7) | 5.1 (1.2, 21.7) | 5.6 (0.9, 13.6 | 0.06 |
| | No | 3 (0.7) | 449 (99.3) | 1 | 1 | - |
| Unsafe sex | Yes | 3 (9.7) | 28 (90.3) | 12 (2.7, 53.4) | 8.2 (1.5,16.7) | 0.003* |
| | No | 5 (0.9) | 567 (99.1) | 1 | 1 | - |
| STI history (women) | Yes | 5 (4.8) | 100 (95.2) | 8.2 (1.9, 35.0) | 4.9 (0.4,55.6) | 0.1 |
| | No | 3 (0.6) | 495 (99.4) | 1 | 1 | - |
| STI history (Husband) | Yes | 3 (6.5) | 43 (93.5) | 9 (1.9, 4.5) | 0.6 (0.4, 9.5) | 0.7 |
| | No | 4 (0.8) | 516 (99.2) | 1 | 1 | - |
| History of incarceration | Yes | 3 (12.0) | 22 (88.0) | 15.6 (3.5, 69.5) | 9.3 (1.6,20.8) | 0.01* |
| | No | 5 (0.9) | 573 (99.1) | 1 | 1 | - |

1: reference category; AOR: Adjusted odds ratio; COR: Crude odds ratio; HBV: Hepatitis B virus; STI: Sexually transmitted infection

* The observed difference is statistically significant

Indonesia (21.6%) [4]. The lower prevalence in this study can be attributed to population deference, high-risk behavioral practices like intravenous drug users (IDU), and homosexuality, which are found to be influential factors in these studies and make the prevalence high. However, HIV/HBV co-infection observed in our study was higher compared to the previous study conducted in the same region, Dessie, Ethiopia (0.5%) [26]. The higher prevalence observed in our study may be attributed to it being conducted across multiple referral hospitals, where women who are aware of their health status might have ANC contact at a referral hospital for comprehensive care. Additionally, our analysis includes all ANC visitors, not just women at initial contact, including those with known infected women, which raises the prevalence even higher. It is also higher than the study in the southwest region of China (0.2%) [27]. This difference might be associated with healthcare-related factors. In developed countries, including China, the standard of screening processes and availability of immunization are higher [3]. In contrast, Ethiopia does not include HBV immunization as a standard practice, which may lead to an increased prevalence of HIV/HBV co-infection in our study setting [1].

If not early diagnosed and treated, HBV and syphilis co-infections can cause severe complications, leading to high rates of morbidity and mortality. In this study, the observed HBV/syphilis co-infection was (0.8%) (95% CI: 0.3, 1.9). It is consistent with the previous studies conducted in Cameroon (0.4%) [28], Nigeria (0.3%) [22], and Angola (0.61%) [17]. Conversely, this finding was lower than the 3.1% reported among HIV-infected individuals in southern Ethiopia [29]. This could be because our study included all pregnant women, regardless of their health status, leading to a lower prevalence due to differences in the denominator. Our finding was also lower compared to the 5.75% prevalence reported in Beijing, China [30].

**Table 5. Bivariate and multivariable logistic regression analysis for identifying risk factors of HBV/Syphilis co-infection among pregnant women in Amhara region referral hospitals, Ethiopia, 2024 (n = 603).**

| Variables | Category | HBV/syphilis | | COR (95% CI) | AOR (95% CI) | P- Value |
|---|---|---|---|---|---|---|
| | | Positive, n (%) | Negative n (%) | | | |
| Surgical history | Yes | 2 (1.9) | 102 (98.1) | 3.2 (0.5, 19.6) | 2.1 (0.3, 14.9) | 0.4 |
| | No | 3 (0.6) | 496 (99.4) | 1 | 1 | - |
| Husband had another sexual partner | Yes | 3 (2.5) | 119 (97.5) | 5.4 (0.9, 32.9) | 2.2 (0.2, 17.4) | 0.4 |
| | No | 2 (0.5) | 432 (99.5) | 1 | 1 | - |
| Use sharp material together | Yes | 3 (2.8) | 105 (97.2) | 7 (1.1, 42.6) | 3.1 (0.3, 29) | 0.3 |
| | No | 2 (0.4) | 493 (99.6) | 1 | 1 | - |
| Unsafe sex | Yes | 1 (3.2) | 30 (96.7) | 4.7 (0.5, 43.6) | 1.4 (0.09, 21) | 0.7 |
| | No | 4 (0.7) | 568 (99.3) | 1 | 1 | - |
| Women had another sexual partner | Yes | 2 (1.9) | 103 (98.1) | 3.2 (0.5, 9.4) | 0.6 (0.05, 7.2) | 0.7 |
| | No | 3 (0.6) | 495 (99.4) | 1 | 1 | - |
| Women with history of STI | Yes | 3 (2.9) | 102 (97.1) | 7.3 (1.2, 44.2) | 6.3 (1.2, 15.9) | 0.03* |
| | No | 2 (0.4) | 496 (99.6) | 1 | 1 | - |
| Alcohol consumption | Yes | 2 (2.0) | 99 (98.0) | 3.5 (0.6, 20.4) | 1.4 (0.2, 14.6) | 0.7 |
| | No | 3 (0.6) | 499 (99.4) | 1 | 1 | - |
| Venous or body piercing for treatment | Yes | 1 (3.1) | 31 (96.9) | 4.5(0.5, 42) | 1.2 (0.9, 17.8) | 0.8 |
| | No | 4 (0.7) | 567 (99.3) | 1 | 1 | - |

1: reference category; AOR: Adjusted odds ratio; COR: Crude odds ratio; HBV: Hepatitis B virus; STI: Sexually transmitted infection

* The observed difference is statistically significant

Discrepancies in prevalence could be attributed to differences in the study populations and variations in reporting periods, reflecting the dynamic nature of the studied infections' epidemic.

Being infected with both HIV and syphilis can make it harder for the immune system to recover. This increases the risk of problems like virologic failure, neurological sequelae, recurrence, and reinfection of syphilis [31]. In the present study, HIV/syphilis co-infection was 1.3% (95 CI: 0.6, 2.6). This is consistent with the studies conducted in Ethiopia in Addis Ababa (1.1%) [9] and the same city at Yirgalem Hospital (2.2%) [18], Indonesia (0.9%) [4], and China (1.8%) [32]. Conversely, it was lower than the studies done in Zambia (40.5%) [14] and Rwanda (27.4%) [13] among HIV infected pregnant women, Angola (6.3%) [17] among pregnant women, western China (18.9%) [25] among HIV-infected pregnant women, and Turkey (5%) [24].

Irrespective of the type of co-infection, the overall prevalence of co-infections in this study was 4.1% (95% CI: 2.7, 6.1), meaning that 4.1% of pregnant women were infected with more than one STI (HIV, HBV, or syphilis). The prevalence of co-infections varies across specific groups and regions. This is due to factors such as sample size, risk status, and reporting period, influenced by changing epidemics of HIV, HBV, and syphilis infection, variations in laboratory procedures and algorithms, as well as variations in the prevention program. Consistent prevalence across different regions underscores the ongoing challenges posed by co-infections [1].

The extent of co-infections can be influenced by factors like the prevalence of each infection in a community, the associated risk behaviors, and the timing of exposure to each infection. In this study, pregnant women with a history of contact with jaundice patients were 5.5 times more likely to be co-infected with HBV and HIV compared to those without such contact. The rationale is that jaundice, characterized by yellowing of the skin and eyes, is a common symptom of acute HBV infection. When a person shows jaundice due to HBV, it signifies a high

level of HBV in their bloodstream, making contact with their blood or other bodily fluids (like saliva or semen) a potential means of transmitting the virus [3]. Women already infected with HIV before or after exposure to jaundiced patients face an increased risk of dual infection due to the potential synergistic effects of multiple infections, which can weaken the immune system and enhance susceptibility to more than one infection [11].

The factor underscores, the importance of implementing WHO recommendations for screening pregnant women for HIV, HBV, and syphilis, along with HBV vaccination for non-immune individuals [3]. Despite HBV being 100 times more infectious than HIV and preventable through vaccination, adult HBV vaccination is not standard practice in Ethiopia [1], leaving pregnant women vulnerable. Identifying a history of contact with a jaundiced patient as a risk factor for HIV/HBV co-infection underscores the urgent need for a national policy to include HBV vaccination as part of comprehensive care.

Pregnant women with a history of unsafe sex were 8.2 times more likely to become infected with HIV/syphilis co-infection than participants without such a history. This finding is supported by a study conducted in southern Ethiopia [29]. It is a fact that safe sex is recommended to prevent the transmission of STIs [1]. Likewise Pregnant women with a history of incarceration were 9.3 times more likely to be susceptible to co-infection with HIV and syphilis than their counterparts. This might be due to confinement-related conditions, socioeconomic disadvantage, substance abuse, and high-risk sexual behaviors within this population before, during, and after incarceration [33]. They might engage in risky behaviors such as injecting drugs, sharing needles and objects such as toothbrushes, razors, or hair clippers, and skin piercings, making disease transmission high [4].

## Limitations of the study

Despite the strengths provided by the multi-center approach, which aims to gather comprehensive data from different geographical locations and significantly enhance the generalizability of the results, the study acknowledges some important limitations. Firstly, the reliance on rapid serological tests for data collection may result in false positives or negatives, potentially leading to an underestimation or overestimation of the study's findings. Secondly, the study depends on self-reported data for variables such as sexual behavior, which introduces the possibility of reporting bias or social desirability bias, thereby potentially compromising the accuracy of the results.

## Conclusion

Based on our findings, we conclude that the seroprevalence of HIV, HBV, and syphilis co-infections among pregnant women in the Amhara region referral hospitals was relatively high as compared to other Ethiopian studies and the 2019 World Health Organization reports. The major factors that affect co-infections are unsafe sexual practices, contact with jaundice patients, incarceration history, and women with a history of STIs. Therefore, to enhance our understanding of the co-infection burden, it is better to conduct further investigations utilizing additional serological markers for HBV, HIV, and syphilis, along with molecular diagnostic methods, given the severity of the co-infection.

## Supporting information

**S1 File.**
(SAV)

## Acknowledgments

We sincerely thank the ANC coordinators, staff, and participants for their essential contributions and assistance. We also express our gratitude to the Amhara region referral hospitals for their support and cooperation throughout the study.

## Author Contributions

**Conceptualization:** Degsew Ewunetie Anteneh, Eden Bishaw Taye, Asmra Tesfahun Seyoum, Alemken Eyayu Abuhay, Endeshaw Admassu cherkose.

**Data curation:** Degsew Ewunetie Anteneh, Eden Bishaw Taye, Asmra Tesfahun Seyoum, Alemken Eyayu Abuhay, Endeshaw Admassu cherkose.

**Formal analysis:** Degsew Ewunetie Anteneh, Eden Bishaw Taye, Asmra Tesfahun Seyoum, Alemken Eyayu Abuhay, Endeshaw Admassu cherkose.

**Funding acquisition:** Degsew Ewunetie Anteneh, Eden Bishaw Taye, Asmra Tesfahun Seyoum, Alemken Eyayu Abuhay, Endeshaw Admassu cherkose.

**Investigation:** Degsew Ewunetie Anteneh, Eden Bishaw Taye, Asmra Tesfahun Seyoum, Alemken Eyayu Abuhay, Endeshaw Admassu cherkose.

**Methodology:** Degsew Ewunetie Anteneh, Eden Bishaw Taye, Asmra Tesfahun Seyoum, Alemken Eyayu Abuhay, Endeshaw Admassu cherkose.

**Project administration:** Degsew Ewunetie Anteneh, Eden Bishaw Taye, Asmra Tesfahun Seyoum, Alemken Eyayu Abuhay, Endeshaw Admassu cherkose.

**Resources:** Degsew Ewunetie Anteneh, Eden Bishaw Taye, Asmra Tesfahun Seyoum, Alemken Eyayu Abuhay, Endeshaw Admassu cherkose.

**Software:** Degsew Ewunetie Anteneh, Eden Bishaw Taye, Asmra Tesfahun Seyoum, Alemken Eyayu Abuhay, Endeshaw Admassu cherkose.

**Supervision:** Degsew Ewunetie Anteneh, Eden Bishaw Taye, Asmra Tesfahun Seyoum, Alemken Eyayu Abuhay, Endeshaw Admassu cherkose.

**Validation:** Degsew Ewunetie Anteneh, Eden Bishaw Taye, Asmra Tesfahun Seyoum, Alemken Eyayu Abuhay, Endeshaw Admassu cherkose.

**Visualization:** Degsew Ewunetie Anteneh, Eden Bishaw Taye, Asmra Tesfahun Seyoum, Alemken Eyayu Abuhay, Endeshaw Admassu cherkose.

**Writing – original draft:** Degsew Ewunetie Anteneh, Eden Bishaw Taye, Asmra Tesfahun Seyoum, Alemken Eyayu Abuhay, Endeshaw Admassu cherkose.

**Writing – review & editing:** Degsew Ewunetie Anteneh, Eden Bishaw Taye, Asmra Tesfahun Seyoum, Alemken Eyayu Abuhay, Endeshaw Admassu cherkose.

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
