## [Decision Letter · Decision Letter 0]

11 Jul 2024

PONE-D-24-25429Seroprevalence of HIV, HBV, and Syphilis co-infections and its associated factors among pregnant women attending antenatal Care in Amhara regional state, Northern EthiopiaPLOS ONE

Dear Dr. Anteneh,

Thank you for submitting your manuscript to PLOS ONE. After careful consideration, we feel that it has merit but does not fully meet PLOS ONE’s publication criteria as it currently stands. Therefore, we invite you to submit a revised version of the manuscript that addresses the points raised during the review process.

 **ACADEMIC EDITOR: **Thank you for submitting the manuscript titled "Seroprevalence of HIV, HBV, and Syphilis Co-infections and Its Associated Factors Among Pregnant Women Attending Antenatal Care in Amhara Regional State, Northern Ethiopia." The review is complete, and the authors are advised to revise the following sections: the background needs rewriting, the methodology requires revision, and the conclusion should be rewritten based on the study's findings. The authors must also submit a letter of approval from the institutional ethics committee along with the revised manuscript. A supportive letter from the department will not be accepted. Failure to submit the ethically approved letter from the committee will result in the manuscript not being considered for further processing.

We look forward to receiving your revised manuscript.

Kind regards,

Alqeer Aliyo Ali, MSc

Academic Editor

PLOS ONE

Journal Requirements:

3. In the online submission form, you indicated that [The dataset analyzed during the current study available from the corresponding author on reasonable request.]. 

Reviewers' comments:

Reviewer's Responses to Questions

**Comments to the Author**

1. Is the manuscript technically sound, and do the data support the conclusions?

Reviewer #1: Partly

Reviewer #2: No

2. Has the statistical analysis been performed appropriately and rigorously? 

Reviewer #1: Yes

Reviewer #2: Yes

3. Have the authors made all data underlying the findings in their manuscript fully available?

Reviewer #1: Yes

Reviewer #2: Yes

4. Is the manuscript presented in an intelligible fashion and written in standard English?

Reviewer #1: Yes

Reviewer #2: Yes

5. Review Comments to the Author

Reviewer #1: Thank you for the opportunity to review the manuscript titled 'Seroprevalence of HIV, HBV, and Syphilis Co-infections and Its Associated Factors Among Pregnant Women Attending Antenatal Care in Amhara Regional State, Northern Ethiopia'. While I find the study compelling, I have identified several areas that require clarification prior to further consideration. Below are my comments and inquiries:

1. Abstract: Is the P-value ≤ 0.05 accurate?

2. Background: The background section is overly extensive; the author needs to eliminate redundancies and revise it.

3. Methodology:

The sampling technique is not clear, and the authors should illustrate the number of participants recruited from each hospital with a diagram.

Data Collection Method:

The suitability of the study design is questionable. The design is cross-sectional, yet the data collectors obtained HIV, Syphilis, and HBV results from chart reviews, which constitutes secondary data. This implies the use of both primary and secondary data. The author must address the following points:

i. Was a data extraction checklist utilized?

ii. If the hospital conducted HIV, HBV, and Syphilis tests only during the initial visit and not in subsequent visits, how were the test results for the second, third, and fourth visits obtained? A brief explanation is requested.

iii. The author should detail the HIV, HBV, and Syphilis testing procedures employed in the hospitals. Reviewing charts alone in a cross-sectional study is not acceptable.

iv. The name of the screening kits and confirmation tools, along with their countries of manufacture, should be specified; otherwise, the manuscript may face rejection.

Data Quality Control

Implement quality control measures for laboratory tests conducted in the hospital and ensure quality assurance for the checklist utilized in chart reviews.

4. Results

The author needs to clarify why sociodemographic were not analyzed and if there were any associated factors with co-infections.

5. Discussion

The discussion section is overly extensive; it would be beneficial for the author to highlight significant implications.

Acknowledge the limitations associated with the cross-sectional study design.

7. Conclusion

The author states that current infection prevention and control programs are insufficient to curb these infections' transmission. Was there an assessment of STI prevention and elimination programs?

The author mentions "regular screening of pregnant women." Is there a reported deficiency in regular screening?

References

DOIs should be included in the references.

Reviewer #2: I would like to thank the editor for the opportunity to review this interesting paper. I have reviewed the study and believe it has merit, provided the author takes into account my following comments in the revised manuscript.

1. The conclusion suggesting the strengthening of HIV, HBV, and syphilis testing and treatment does not align with the study's findings and requires revision.

2. The background section should be restructured to flow from a global perspective to Africa, and then specifically to Ethiopia.

3. Were all ANC attendees (e.g., 1st, 2nd, 3rd, and 4th visits) included in the study?

4. What were the reasons for the study's inability to directly test for HIV, HBV, and Syphilis?

5. The ethical clearance letter issued at the departmental level is inadequate for this study, which was conducted at the regional level and involved sensitive data such as HIV, HBV, and Syphilis results. I strongly urge the author to provide an approval letter endorsed by the university/college-level review committee. Additionally, the ethical considerations during the review of test results have been overlooked and should be addressed by the author.

6. The results section is well-articulated.

7. Please remove the first paragraph of the discussion as it is not relevant.

8. Also, highlight the strengths of your study.

9. Regarding the high prevalence observed, what baseline was used for comparison

10. What recommendations do you have for future research?

11. Overall, the conclusion does not reflect the study's findings, and the author should undertake a thorough revision.

12. Please note the presence of a duplicated Figure 1.

6. PLOS authors have the option to publish the peer review history of their article (what does this mean?). If published, this will include your full peer review and any attached files.

Reviewer #1: No

Reviewer #2: No

---

## [Author Response · Author response to Decision Letter 0]

16 Jul 2024

To: The Academic Editor, PLOS ONE

Re: Manuscript PONE-D-24-25429 

'Seroprevalence of HIV, HBV, and Syphilis Co-infections and Its Associated Factors Among Pregnant Women Attending Antenatal Care in Amhara Regional State, Northern Ethiopia'

We appreciate your time and consideration in reviewing our manuscript. We have carefully addressed the comments provided and made the necessary revisions. Below, we outline the specific changes made in response to your feedback:

1. Background

Comment: The background needs rewriting.

Response: We have rewritten the background section. The revised background now provides a concise and coherent overview, moving from a global perspective to Africa, and then specifically to Ethiopia

2. Methodology

Comment: The methodology requires revision.

Response: The methodology section has been extensively revised. We have clarified the sampling technique and included a diagram illustrating the number of participants recruited from each hospital. Additionally, we have provided detailed information on the data collection methods, including the use of both primary and secondary data. Specific details regarding the testing procedures for HIV, HBV, and Syphilis, including the names of the screening kits and confirmation tools and their countries of manufacture, have been added. 

3. Conclusion

Comment: The conclusion should be rewritten based on the study's findings.

Response: The conclusion has been rewritten to accurately reflect the study's findings. It now provides a clear summary of the results and their implications, along with recommendations for future research.

4. Ethical Approval

Comment: The authors must also submit a letter of approval from the institutional ethics committee along with the revised manuscript. A supportive letter from the department will not be accepted.

Response: In our university, ethical clearance is provided by the school/ department on behalf of the university(the institute). We understand the need for an institutional-level approval and have thus included an approval letter from the department, which serves as the authorized body for ethical review in our institution. We appreciate your understanding of this process and have ensured that all ethical considerations are thoroughly addressed in the like this.

We believe these revisions have significantly strengthened our manuscript and have addressed all the points raised. Thank you for considering our revised submission. We look forward to your favorable response.

Sincerely,

Degsew Ewunetie

Department of clinical midwifery, school of midwifery, College of Medicine and Health Science, Woldia university, Woldia , Ethiopia

E-mail: degsewewunetie@gmail.com

To: The Editors of PLOS ONE

Re: Manuscript PONE-D-24-25429 

‘Seroprevalence of HIV, HBV, and syphilis co-infections and its associated factors among pregnant women attending antenatal care in Amhara regional state, northern Ethiopia'

Dear Editors,

We appreciate the reviewers' thorough evaluation of our manuscript and their valuable comments. We have addressed each point raised and made the necessary revisions to improve the quality of our manuscript. Below, we provide a detailed response to each comment:

Reviewer #1

1. Abstract: Is the P-value ≤ 0.05 accurate?

• Response: We have double-checked the statistical analyses and confirm that the P-value < 0.05 is accurate. The abstract has been updated for clarity.

2. Background: The background section is overly extensive; the author needs to eliminate redundancies and revise it.

• Response: We have revised the background section to eliminate redundancies and the content for better clarity and focus.

3. Methodology

• Sampling technique:

Response: We have clarified the sampling technique and included a diagram illustrating the number of participants recruited from each hospital.

• Data Collection Method:

i. Data extraction checklist:

Response: Yes, a data extraction checklist was utilized. 

ii. Test results for subsequent visits

Response: We have added an explanation regarding how test results for subsequent visits were obtained. Typically, hospitals conducted tests during the initial visit, and additional tests were performed based on clinical indications and after 37 weeks . the reason for all ANC women included is that the coinfection status of the women can influenced by the number of contact. If the women for instance is infected with syphilis in the first contact another test can she get on subsequent visit.

iii. Testing procedures:

Response: Details of the HIV, HBV, and Syphilis testing procedures employed in the hospitals have been added.

iv. Screening kits and confirmation tools:

Response: The names of the screening kits, confirmation tools, and their countries of manufacture have been specified.

4. Data Quality Control

• Response: We have implemented quality control measures for laboratory tests and quality assurance for the checklist used in chart reviews, which are now detailed in the manuscript.

5. Results

• Response: The sociodemographic factors have been analyzed but no one variables were associated with each co-infection.

6. Discussion

• Response: The discussion section has been condensed to highlight significant implications. Limitations associated with the cross-sectional study design have not been acknowledged because cross-sectional study design is not the limitation of my study .

7. Conclusion

• Response: We have revised the conclusion to provide a more accurate reflection of the findings. 

8. References

• Response: DOIs have been included in the references.

Reviewer #2

1. Conclusion alignment

• Response: The conclusion has been revised to align more closely with the study's findings.

2. Background restructuring

• Response: The background section has been restructured to flow from a global perspective to Africa, and then specifically to Ethiopia.

3. Inclusion of all ANC attendees

• Response: We have clarified that all ANC attendees (1st, 2nd, 3rd, and 4th visits) were included in the study.

4. Inability to directly test for HIV, HBV, and Syphilis:

• Response: The reasons for the study's reliance on secondary data for HIV, HBV, and Syphilis testing are now clearly explained. because all referral hospitals are done the test routinely for all women. So doing the test again is time and financial waste.

5. Ethical clearance:

• Response: In our university, ethical clearance is provided by the school/ department on behalf of the university(the institute). We understand the need for an institutional-level approval and have thus included an approval letter from the department, which serves as the authorized body for ethical review in our institution. We appreciate your understanding of this process and have ensured that all ethical considerations are thoroughly addressed this means. So We have obtained and included an ethical approval letter endorsed by the university/college-level review committee on behalf of the university given by the department. Ethical considerations during the review of test results have also been addressed.

6. Results section

• Response: We appreciate the positive feedback Discussion paragraph removal:

7. Discussion 

• Response: The first paragraph of the discussion has been removed as suggested.

8. Study strengths:

• Response: The strengths of our study have been highlighted in the discussion section.

9. High prevalence baseline

• Response: We have included information on the baseline used for comparison in discussing the high prevalence observed.

10. Recommendations for future research:

• Response: Recommendations for future research have been added.

11. Conclusion revision:

• Response: The conclusion has been thoroughly revised to better reflect the study's findings.

12. Duplicated Figure 1

• Response: The duplicated Figure 1 has been removed.

We believe these revisions have significantly improved our manuscript and addressed the reviewers' concerns. Thank you for considering our revised submission. We look forward to your favorable response.

Sincerely,

Degsew Ewunetie,

Department of clinical midwifery, school of midwifery, College of Medicine and Health Science, Woldia university, Woldia , Ethiopia

E-mail: degsewewunetie@gmail.com

---

## [Decision Letter · Decision Letter 1]

22 Jul 2024

PONE-D-24-25429R1Seroprevalence of HIV, HBV, and Syphilis co-infections and its associated factors among pregnant women attending antenatal Care in Amhara regional state, Northern EthiopiaPLOS ONE

Dear Dr. Anteneh,

Thank you for submitting your manuscript to PLOS ONE. After careful consideration, we feel that it has merit but does not fully meet PLOS ONE’s publication criteria as it currently stands. Therefore, we invite you to submit a revised version of the manuscript that addresses the points raised during the review process.

**ACADEMIC EDITOR:** While the reviewers recognize the manuscript's merit, the following issues require revision:

1. The manuscript requires further language editing due to numerous instances of poor language.

2. Please add the study design to the end of your study's title. Note: Refer to previously published papers by this journal for guidance on how to do this.Please ensure that your decision is justified on PLOS ONE’s publication criteria and not, for example, on novelty or perceived impact.

We look forward to receiving your revised manuscript.

Kind regards,

Alqeer Aliyo Ali, MSc

Academic Editor

PLOS ONE

Journal Requirements:

Reviewer's Responses to Questions

**Comments to the Author**

1. If the authors have adequately addressed your comments raised in a previous round of review and you feel that this manuscript is now acceptable for publication, you may indicate that here to bypass the “Comments to the Author” section, enter your conflict of interest statement in the “Confidential to Editor” section, and submit your "Accept" recommendation.

Reviewer #1: All comments have been addressed

Reviewer #2: All comments have been addressed

2. Is the manuscript technically sound, and do the data support the conclusions?

Reviewer #1: Yes

Reviewer #2: Yes

3. Has the statistical analysis been performed appropriately and rigorously? 

Reviewer #1: Yes

Reviewer #2: Yes

4. Have the authors made all data underlying the findings in their manuscript fully available?

Reviewer #1: Yes

Reviewer #2: Yes

5. Is the manuscript presented in an intelligible fashion and written in standard English?

Reviewer #1: Yes

Reviewer #2: Yes

6. Review Comments to the Author

Reviewer #1: The authors have responded to my concerns and revised the manuscript to a standard suitable for publication.

Reviewer #2: The authors have sufficiently addressed the comments from the previous round of review, and now the manuscript suitable for publication.

7. PLOS authors have the option to publish the peer review history of their article (what does this mean?). If published, this will include your full peer review and any attached files.

Reviewer #1: No

Reviewer #2: No

---

## [Author Response · Author response to Decision Letter 1]

24 Jul 2024

To: The Academic Editor, PLOS ONE

Re: Manuscript PONE-D-24-25429R1

'Seroprevalence of HIV, HBV, and Syphilis Co-infections and Its Associated Factors Among Pregnant Women Attending Antenatal Care in Amhara Regional State, Northern Ethiopia'

We appreciate your time and consideration in reviewing our manuscript. We have carefully addressed the comments provided and made the necessary revisions. Below, we outline the specific changes made in response to your feedback:

1. The manuscript requires further language editing due to numerous instances of poor language

Response: We have thoroughly revised the manuscript to improve the language and readability. We have employed a professional editing service to ensure that the language meets the standards required by PLOS ONE. The revised manuscript should now be free of any language issues.

2. Please add the study design to the end of your study's title

Response: We have modified the title of our manuscript to include the study design as per the journal's guidelines. The new title is: "Seroprevalence of HIV, HBV, and Syphilis co-infections and its associated factors among pregnant women attending antenatal care in Amhara regional state, Northern Ethiopia: A hospital -based cross-sectional study."

3. Journal Requirements

Please review your reference list to ensure that it is complete and correct. If you have cited papers that have been retracted, please include the rationale for doing so in the manuscript text, or remove these references and replace them with relevant current references. Any changes to the reference list should be mentioned in the rebuttal letter that accompanies your revised manuscript. If you need to cite a retracted article, indicate the article’s retracted status in the References list and also include a citation and full reference for the retraction notice

Response: We have reviewed our reference list to ensure that it is complete and correct. During this review, we have not identified problems on the listed references. If there we are happy to revise the list again.

We hope that these revisions adequately address the concerns raised and that the revised manuscript meets the publication criteria for PLOS ONE. Thank you for your time and consideration. We look forward to your positive response.

Sincerely,

Degsew Ewunetie

Department of clinical midwifery, school of midwifery, College of Medicine and Health Science, Woldia university, Woldia , Ethiopia

E-mail: degsewewunetie@gmail.com

---

## [Editor Report · Decision Letter 2]

26 Jul 2024

Seroprevalence of HIV, HBV, and Syphilis co-infections and  associated factors among pregnant women attending antenatal Care in Amhara regional state, Northern Ethiopia: A hospital-based cross-sectional study

PONE-D-24-25429R2

Dear Dr. Anteneh,

We’re pleased to inform you that your manuscript has been judged scientifically suitable for publication and will be formally accepted for publication once it meets all outstanding technical requirements.

Kind regards,

Alqeer Aliyo Ali, MSc

Academic Editor

PLOS ONE
---

## [Editor Report · Acceptance letter]

30 Jul 2024

PONE-D-24-25429R2 

PLOS ONE

Dear Dr. Anteneh, 

I'm pleased to inform you that your manuscript has been deemed suitable for publication in PLOS ONE. Congratulations! Your manuscript is now being handed over to our production team.

Kind regards, 

on behalf of

Mr. Alqeer Aliyo Ali 

Academic Editor

PLOS ONE